# Synthesis of Superheat-Resistant Polyimides with Enhanced Dielectric Constant by Introduction of Cu(ΙΙ)-Coordination

**DOI:** 10.3390/polym12020442

**Published:** 2020-02-13

**Authors:** Guangtao Qian, Mengjie Hu, Shangying Zhang, Mengxia Wang, Chunhai Chen, Jianan Yao

**Affiliations:** Center for Advanced Low-Dimension Materials, State Key Laboratory for Modification of Chemical Fibers and Polymer Materials, College of Material Science and Engineering, Donghua University, Shanghai 201620, China; 1179133@mail.dhu.edu.cn (G.Q.); 2180373@mail.dhu.edu.cn (M.H.); 2180257@mail.edu.edu.cn (M.W.); cch@dhu.edu.cn (C.C.)

**Keywords:** polyimide, metal coordination crosslinking, high performance

## Abstract

To achieve polyimide-metal complexes with enhanced properties, 5-amine-2-(5-aminopyridin-2-yl)-1-methyl-benzimidazole (PyMePABZ) that contains stiff 2-(2′-pyridyl)benzimidazole (PyBI) was synthesized and exploited to construct the Cu(ΙΙ)-crosslinked polyimides (Cu-PIs). These Cu-PIs exhibited higher dielectric, thermal, and mechanical properties with an increase in Cu^2+^ content. Among them, their dielectric constants (ε_rS_) were up to 43% superior to that of the neat PI, glass transition temperatures (T_g_s) were all over 400 °C, and 5% weight loss temperature (T_5%_) maintained beyond 500 °C. These data indicate that the metal coordination crosslinking provided a useful guide to develop high performance PIs which possess potential application as useful high temperature capacitors.

## 1. Introduction

Polymer film capacitors have been attractive in many research efforts for recent years owing to their low cost, self-healing nature, graceful failure, and flexibility [1,2,3]. The ideal polymer dielectrics normally possess high dielectric constant (ε_r_ > 3, if possible, higher ε_r_ is desirable), such as polyester (PET), polyethylene naphthalate (PEN), and poly(vinylidene fluoride) (PVDF) [4,5]. Unfortunately, their low thermal stability is the main restriction for the application at high operating temperature, because the dielectric properties decrease dramatically near or above T_g_ [6,7]. Thus, to increase the operation temperature, initial interest in new dielectric materials is generally directed to enhance dielectric constants with better thermal durability.

PIs with excellent thermal and mechanical properties have been studied as a kind of high-performance engineering polymer [8,9,10]. It is one of the most promising polymers with outstanding T_g_s, nevertheless, their dielectric constant is typically ~3.5, which limits their use as high dielectric materials [11,12]. Hence, various modified PIs have been widely studied, generally including a mixture with high dielectric constant fillers and modifications of molecular architecture [7,13,14,15]. Although these successful methodologies led to a great increase of dielectric properties, the former causes the degradation of mechanical properties owing to the phase segregation of fillers and the latter brings the reduced thermal properties with the employment of flexible blocks (e.g., ether linkage). Differently, the supramolecular polymers that exploit metal–ligand interactions, possess high dielectric constant because of the improved permanent dipoles caused by metal atoms [16,17], and the metal-crosslinking can further improve their thermal and mechanical properties [17,18].

In previous researches, PyBI possesses an NH unit that can be substituted with appropriate groups and two lone electron pairs of N atoms that can chelate to a metal center [19,20,21]. To introduce the special structure into PIs, PyMePABZ was designed and synthesized as a novel diamine (Scheme 1). The substituent of the proton in the NH eliminates the effects of hydrogen atom which has intramolecular H-bonding ability [22,23]. In addition, the copolyimide (neat PI) was prepared by copolycondensation with PyMePABZ and 4,4’-oxydianiline (ODA). The Cu(ΙΙ)-crosslinked polyimides (Cu-PIs) with different content of the metal cation were prepared (Scheme 2) and utilized to discuss the relationship between metal–ligand interactions and properties of polymers.

## 2. Experimental

### 2.1. Materials

N-methyl-4-nitrobenzene-1,2-diamine and 5-nitropicolinoyl chloride were purchased and used as received from Aladdin Chemical Co. Ltd. ODA, pyromellitic dianhydride (PMDA), triethylamine (Et_3_N), p-Toluenesulfonic acid (p-TSA), Cupric(II) acetylacetonate (Cu(C_5_H_7_O_2_)_2_), 80% hydrazine monohydrate, 10% palladium on charcoal and other reagents were purchased from Sinopharm Chemical Reagent Beijing Co. Ltd. (Beijing, China). N-methylpyrolidon (NMP), gamma-butyrolactone (GBL), and dioxane were purified by vacuum distillation and stored over 4 Å molecular sieves prior to use. All the other commercially available solvents and reagents were purchased from Sigma-Aldrich (Shanghai, China), used without further purification.

### 2.2. Measurements

^1^H NMR spectra were obtained on a Bruker 600 AVANCE III spectrometer (Billerica, MA, USA), in which dimethyl sulfoxide-*d*_6_ was used as a solvent. Elemental analyses were carried out by Elmentar Vario EL-III (Hanau, Germany). The inherent viscosities (η_inhs_) were measured with an Ubbelohde viscometer at 25 ± 0.1 °C using NMP as a solvent. Fourier transform infrared (FTIR) spectra were conducted on a ThermoFisher Nicolet 6700 infrared spectrometer (Waltham, MA, USA) by averaging 32 scans within the range of 4000–400 cm^−1^, with the sample form of powder (monomers) and thin polymer films (~5 um). The thermogravimetric analysis (TGA) was assessed using TA Instrument Discovery TGA 550 (New Castle, DE, USA) with a constant heating rate of 10 °C min^−1^ under nitrogen. Dynamic mechanical analysis (DMA) was performed on a TA Instrument DMA Q800 (New Castle, Delaware, USA) with a heating rate of 5 °C min^−1^ and a frequency of 1 Hz. Mechanical properties of polymer membranes were evaluated on a Shimadzu AG-I universal testing apparatus (Kyoto, Japan) at speed of 5 mm min^−1^, and tensile modulus (E), tensile strength (σ), and elongation at break (ε) were demonstrated as the average of five strips. The X-ray photoelectron spectroscopy (XPS) spectrum of the sample was achieved through a ThermoFisher Escalab 250Xi system (Waltham, MA, USA) equipped with Mg anode. Dielectric constants (ε_r_) and dissipation factors (Tan δ) were measured on a precision LCR meter Agilent E4980A (Santa Clara, CA, USA) equipped with dielectric test fixture 16451B at frequencies of 20Hz-1MHz, using the equation:(1)$$\varepsilon_{r}=\frac{t_{m}\times C_{p}}{\pi {\left(\frac{d}{2}\right)}^{2}\times \varepsilon_{o}}$$

Among them, t_m_ was average film thickness [m], C_p_ was sample capacitance [F], d was diameter of thin-film electrode [m] (the sprayed gold was served as electrodes) and ε_o_ is a constant of 8.854 × 10^−12^ F/m (dielectric constant in vacuum).

### 2.3. Synthesis of Monomers

#### 2.3.1. Synthesis of 5-nitro-2-(5-nitropyridin-2-yl)-1-methyl-benzimidazole (3)

A 250-mL flask is charged with N-methyl-4-nitrobenzene-1,2-diamine (10.0 g, 0.060 mol), Et_3_N (7.3 g, 0.072 mol), and THF (100.0 mL). 5-nitropicolinoyl chloride (13.4 g, 0.072 mol) was added dropwise at a rate maintaining temperature of the reaction mixture being below 10 °C. After the mixture was stirred at room temperature overnight, it was poured in 200.0 mL of water, and the suspension was collected by filtration. The crude product was dried and then dissolved in GBL (200.0 mL) containing p-TSA (12.4 g, 0.072 mol). The reaction mixture was allowed to warm to 190 °C and stirred for 5 h. Upon completion of the reaction, the obtained mixture was cooled to room temperature and slowly poured into a large quantity ice water. The precipitated solid was filtered under vacuum and the filter cake is washed with 100 mL of ethanol. After evaporation in vacuum and recrystallization from DMSO, a resulting yellow solid was obtained (12.7 g, two-step yield: 71.0%). Melting point: 264–265 °C. ^1^H NMR (DMSO-*d*_6_, ppm): *δ* = 9.55 (s, 1 H), 8.81 (d, 1 H, *J* = 8.6 Hz), 8.69 (s, 1 H), 8.62 (d, 1 H, *J* = 8.6 Hz), 8.29 (d, 1 H, *J* = 8.6 Hz), 7.98 (d, 1 H, *J* = 8.9 Hz), 4.34 (s, 3 H). FTIR (KBr, *ν*, cm^−1^): 1615, 1595 (C=N/C=C stretching of ring), 1523, 1326 (NO_2_ asymmetric and symmetric stretching). Anal. Calcd. for C_13_H_9_N_5_O_4_: C, 52.18; H, 3.03; N, 23.40. Found: C, 51.93; H, 3.21; N, 23.67.

#### 2.3.2. Synthesis of 5-amine-2-(5-aminopyridin-2-yl)-1-methyl-benzimidazole (PyMePABZ)

A mixture of 3 (10.0 g, 0.033 mol), Pd/C (1.0 g), and EtOH (100 mL) was stirred and refluxed for 1 h. Then, hydrazine hydrate (80%, 25 mL) was dropwise added to the mixture for further stirring for 5 h at 80 °C. Upon completion of the reaction, addition of water gave a brown powder which was collected and washed with a mixture of EtOH/H_2_O. The obtained solid was purified by recrystallizing from a mixture of EtOH/H_2_O (5:1) to give the desired product (7.4 g, yield: 93.1%). Melting point: 236–237 °C. ^1^H NMR (DMSO-*d*_6_, ppm): *δ* = 8.02 (d, 1 H, *J* = 2.7 Hz), 7.92 (d, 1 H, *J* = 8.6 Hz), 7.19 (d, 1 H, *J* = 8.5 Hz), 7.04 (dd, 1 H, *J* = 8.6, 2.7 Hz), 6.75 (d, 1 H, *J* = 1.9 Hz), 6.60 (dd, 1 H, *J* = 8.5, 1.9 Hz), 5.76 (s, 2 H), 4.74 (s, 2 H), 4.06 (s, 3 H). FTIR (KBr, *ν*, cm^−1^): 1624, 1588 (C=N/C=C stretching of ring), 3418, 3294, 3172 (amine NH). Anal. Calcd. for C_13_H_13_N_5_: C, 65.25; H, 5.48; N, 29.27. Found: C, 65.36; H, 5.62; N, 29.01.

### 2.4. Preparation of Polymers

Cu-PIs were prepared through a three-stage synthesis, including poly(amic acid) (PAA) precursors, Cu(ΙΙ)-PAAs, and a thermal imidization process (Scheme 2). First, after the PyMePABZ (0.2393 g, 1.0 mmol) and ODA (0.6007 g, 3.0 mmol) were completely dissolved in NMP (15.4 g), PMDA (0.8725 g, 4.0 mmol) was slowly added with continuous stirring at room temperature for 16 h. The viscous PAA solution generally maintained a solid content of 10 wt%. Second, anhydrous Cu(C_5_H_7_O_2_)_2_ (0, 0.0524, 0.0916, 0.1309 g) was separately added into the sealed bottle containing PAAs and stirred for another 16 h at room temperature. Finally, the Cu(ΙΙ)-PAA solutions were cast on a clean and dry glass plate, and cured in an oven with a curing program typically at 80, 150, 250, and 350 °C for 1 h at each temperature. The thickness of the PI films was approximately 20 μm. All PIs were divided into two series: Neat-PI and Cu-PIs.

## 3. Results and Discussion

### 3.1. Molecular Structure and Polymerization Characterization

As illustrated in Scheme 1, PyMePABZ was synthesized from the reaction sequence. In the FTIR spectrum (Appendix A), the characteristic absorptions (3418–3172 cm^−1^) arose from -NH_2_ stretching bands that could be found obviously in PyMePABZ, also, the results of ^1^H NMR (Figure 1) showed two obvious signals of amine proton located at near 5.76 and 4.74 ppm corresponding to the amine protons which were linked to benzimidazole (BI) and pyridine ring, respectively. These data indicated the diamine was synthesized successfully. Additionally, the signals of methyl (4.06 ppm) and pyridine (8.02–7.04 ppm) illustrated the successful introduction of the corresponding units.

The inherent viscosities of the Gu-PAAs were in the range of 1.10–0.83 dL g^−1^ (Table 1), which guaranteed the successful formation of the tough and flexible PI films. Interestingly, the values showed an upward trend when the Cu(ΙΙ) content increased. This could be ascribed to the appearance of the crosslinked system caused by the Cu^2+^ in the PAA solutions, and the degree of crosslinking increased with the increase of Cu^2+^ content. The PyBI ligand could coordinate with Cu ions to form 2:1 complexes, therefore, 50% Cu-PAA (0.5 equiv. Cu per PyMePABZ) showed the highest η_inh_. In this work, higher proportions of PyMePABZ and Cu (ΙΙ) content (e.g., n:m = 50:50 and 0.5 equiv. Cu per PyMePABZ) in the metal-crosslinked PAAs had been achieved, but the mixture generated gels and lost the fluidity, resulting in weak film-forming capability. The phenomenon was herein assumed to correspond to the insolubility of the stiff crosslinked unit (PyBI-Cu^2+^) when it reached a certain degree [18].

Neat PI and Cu-PIs were prepared according to Scheme 2. The molecule structures of the successfully prepared PIs were investigated by FTIR as shown in Figure 2a. The typical characteristic peaks at around 1777, 1723, 1376 cm^−1^ were assigned to imide carbonyl asymmetric stretching, imide carbonyl symmetric stretching, and cyclic C–N stretching, indicating the successful formation of imide rings. The breathing peak of BI ring (1307 cm^−1^) and band of the pyridine ring (~1600 cm^−1^) suggested the corresponding groups had been introduced into the PI chains [24,25,26]. Moreover, the peak corresponding to the C=O in -CONH- (1660 cm^−1^, gray band domain) was hardly observed, indicating that the imidization of the PAA solutions was fully completed [27,28].

### 3.2. Cu(II)-Coordination of PIs

The formation of Cu-coordination bonds was confirmed by FTIR spectra (Figure 2a) and X-ray photoelectron spectroscopy (XPS) (Figure 2b). As we know, the characteristic band of free pyridine ring was at 1596 cm^−1^ and the band at 1603 cm^−1^ could be defined as the absorption from boned pyridine ring in the polymer FTIR spectra [25,26]. In the resulting PIs, not only the neat PI (no Cu^2+^) had a clear band corresponding to the pure pyridine ring absorption, but also the band assigned to the boned pyridine ring could be observed in the 50% Cu-PI (ligand fully chelate to metal). Additionally, the band position of pyridine ring (~1600 cm^−1^) gradually generated a conspicuous shift after the incorporation of Cu^2+^ into PIs (Figure 2a), which demonstrated the increased amount of complexed pyridine units with an increase in Cu^2+^ content. This was consistent with the results of the data in the inherent viscosities analyses (Table 1). In the further analysis of XPS at the Cu 2p level, all Gu-PIs displayed the Cu 2p_3/2_ main peak (933.1–933.6 eV) and a lower shake-up peak (939–944 eV), which was in agreement with two major XPS characteristics of CuO [29,30]. These results indicated that the Cu ions of the metal-crosslinked PIs were in the 2^+^ oxidation state.

### 3.3. Physical Properties of PIs

The data of dielectric constants was summarized in Figure 3a and Table 1. All the Cu-PIs possessed high values (ε_r_ = 4.5−5.3), which were superior to common commercial PIs. Such high dielectric constants were achieved because of the special complex molecular structure from the ligand PyBI and metal-center Cu(ΙΙ). The complexation, as a kind of highly polarized units, occurred by the d-π bonds between the unoccupied d-orbitals of Cu(II) and the lone electron pairs of N atoms in PyBI segments. Incorporation of the metal-bonded fraction enhanced the orientational polarizability in the resulting PIs and, thus, improved their dielectric constant. This trend was further confirmed by the PIs with higher content of Cu-bonded blocks having ε_r_ higher than that containing less Cu^2+^. Among them, the value of 50% Cu-PI increased 43% than that of the one without Cu-complex. Moreover, the dielectric constant was found to be dependent on the frequency of external field. The values slightly decreased as the frequency increased, because alternating in dipoles could not keep up with changed electric field, resulting in a slight decrease in the orientational polarization at high frequency. As shown in Figure 3b and Table 1, the dielectric loss of PIs prepared here was below 1% at 1 kHz and room temperature. Although the dielectric loss slightly increased at higher frequency or with increasing amount of Cu-complex because of the enhanced dipolar polarization, the values remained below 3.5%, comparable to that of most previously reported polymers for energy storage applications [31,32,33].

Thermal stability of these PI films was evaluated by TGA (Figure 4 and Table 1). When the PIs with side groups endured a high temperature, the beginning weight loss was considered to be related to the thermal degradation of the methyl side group [34,35]. In the resulting Cu-PIs, the efficient crosslinking reactions facilitated the possibility of methyl rupture and, thus the degradation behavior slightly shifted to low temperatures after the addition of Cu^2+^ in the polymers. However, the crosslinked networks might counteract thermal fragmentation without volatile compounds formation, and that was with crosslinking reactions, the characteristic weight loss of the Cu-PIs showed a more moderate trend [36], which could be clearly recognized in Figure 4. On the other hand, the use of PyBI as a stiff unit increased the π-electron system, which led to a rapid conduction of heat between chemical bonds and alleviated the polymer degradation in a high temperature [37]. Therefore, the observed T_5%_ of the Cu-PIs under a nitrogen atmosphere maintained beyond 500 °C. Further, more metal-crosslinking might reserve Cu^2+^ in the materials to a considerable extent and more residues could be left. The experimental value of Cu-PIs at 800 °C was higher than that of the neat one.

The dielectric constants were believed to increase dramatically above T_g_, resulting in negative effect on the energy storage of polar polymers [7]. Therefore, a high T_g_ was required urgently to increase the operation temperature. In this work, the T_g_s were measured by DMA (Figure 5) and the values of Cu-PIs (Table 1) maintained >400 °C, which were impressive for the polymers used in dielectric capacitors. The more attractive values were largely attributed to the PI substrates with high heat resistance that resulted from the rigid rod-like backbone and enhanced electron donation-acceptance interaction of BI diamines [38,39]. Additionally, the interchain interaction played a key role in influencing T_g_ values for this kind of polymer. The metal-crosslinking, as the strong physical interaction, significantly limited the rotational freedom around single bonds, improving the T_g_. This interaction was further strengthened when the content of Cu^2+^ in the metal-crosslinked PIs increased and, consequently, 50% Cu-PIs had a high T_g_ up to 414 °C.

In addition to improving the thermal performance, the crosslinking effect also enhanced the mechanical properties of these PIs. In the results of Appendix A and Table 1, the Cu-PIs showed the good tensile strength of 137–160 MPa and high elongation at break of 8.2%–9.3%, illustrating their toughness and flexibility. Overall, after the coordination was formed between the Cu^2+^ and PyBI, the intermolecular forces could be increased, leading to the increasing of the PI’s mechanical properties. In particular, 50% Cu-PIs led to an increase of greater than 20% in tensile strength and elongation at break for the neat PIs. While the modulus of these PIs showed little noticeable improvement with the increasing amount of Cu^2+^, the values (~5.8 GPa) would be sufficient for further applications in the capacitor fields.

## 4. Conclusions

A novel diamine MePyPABZ was synthesized and exploited to copolycondensed with ODPA, and the copoly(amic acid) was added to different amounts of Cu^2+^ to prepare the Cu-PIs with imidization processes. The introduction of Cu^2+^ with the formation of metal-complex could increase the polarizability, providing the resulting PIs with higher dielectric constants depending on the Cu^2+^ content. All Cu-PIs exhibited outstanding physical properties, including excellent T_g_ up to 414 °C, high T_5%_ over 500 °C, good tensile strength from 137 to 160 MPa, and elongation at break beyond 8.2%, because of the additional metal-crosslinking. In summary, the metal coordination network in these PIs not only improved their dielectric properties, but also enabled their thermal and mechanical properties to be enhanced, which provided a useful guide to provide high performance polymers for energy storage applications used in high temperature.

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
