# Peer review of "Synthesis of Superheat-Resistant Polyimides with Enhanced Dielectric Constant by Introduction of Cu(ΙΙ)-Coordination"

_polymers, 2020, doi:10.3390/polym12020442_

Round 1

Reviewer 1 Report

Comments for the Authors polymers-696467:

This paper describes the preparation and characterization of the polyimide-metal complexes by incorporating the diamine of 5-amine-2-(5-aminopyridin-2-yl)-1-methyl-benzimidazole (PyMePABZ) to exploit the Cu(II)-crosslinked polyimides (Cu-PIs). These resulted Cu-PIs exhibited enhanced dielectric, thermal and mechanical properties with an increase in Cu2+ content. This approach is interesting and could provide a model to prepare dielectric polymers with potential applications, and the results are also reasonable to support the author's argument. Thus, I recommend the manuscript to be published in this journal after some revisions.

1). Melting points of the prepared diamine and intermideate compoundd should be denoted with temperature range not just an exact value. Foe example, the melting point of diamine PyMePABZ should be expressed 236-237 °C, not 236.2 °C.

2). Regarding the preparation of polymer in section 2.4, why only 4.0 mmol of PMDA dianhydride was added but not 0.5 mmol?

Reviewer 2 Report

The manuscript submitted by Yao et al. reported on the synthesis of polyimides with enhanced dielectric constant by introduction of Cu(II)-coordination. Even if the authors reported sufficiently good experimental data the manuscript in not suitable for the publication in the present way and require substantial changes.  The synthesis is not well described and I suggest to rewrite the paragraph 3.1. in detail and also paragraph 3.2.

Other correction:

- the first time you introduce an abbreviation you have to introduce what means, e.g. ODA on row 46

-scheme 1a is not clear, please improve it.

-experimental results are anticipated on figure 1 b and c; please shit them in the results and discussion section.  

-the sentence on rows 113-115 is not well written, please adjust it.

-the “functionally tectonic units” on row 130  is not the appropriate terminology for this context.

-row 133: Gu-PAA refers probably to Cu-PAA, in addition you refer to Cu-PAA as precursors while on raw 113 the precursor was indicated as PAA. Please, correct and use the appropriate term.

-row 139: “higher proportion of….”  Higher with respect to? Add a reference.

-row 146 “PI and Cu-PIs were prepared according to Fig 1a”; it is not clear in figure 1a the reaction scheme and in this section you have to explain how you prepare them.

-paragraph 3.2 must to be rewrite, it contains refuses probably and is not clear.  E.g. containing content….row 161; or “That suggested”, at the beginning of a sentence is not correct.

-caption on figure 3 refers to a and b, please insert a and b in the figure.

Moreover, when the mouse move over figures 1, 2, and 3 appear the sentence :manuscript European polymer journal that seems you have already submitted the manuscript to this journal and it seems unprofessional.

Round 2

Reviewer 2 Report

-